# The Coupling Coordination Degree of Economic, Social and Ecological Resilience of Urban Agglomerations in China

**DOI:** 10.3390/ijerph20010413

**Published:** 2022-12-27

**Authors:** Xin Xu, Meimei Wang, Mingfeng Wang, Yongchun Yang, Yuliang Wang

**Affiliations:** 1School of Resources and Environment, Lanzhou University, Lanzhou 730000, China; 2School of Urban and Regional Sciences, East China Normal University, Shanghai 200241, China

**Keywords:** urban agglomeration, economic resilience, social resilience, ecological resilience, coupling coordination degree, fuzzy logic method

## Abstract

This paper refines the fuzzy logic method, while constructing a theoretical model of the relationship between economic resilience, social resilience and ecological resilience, and evaluates the coupling coordination between the economic-social-ecological resilience of 197 prefecture-level cities in China’s urban agglomerations in 2019. Findings include: (1) The mean ecological resilience of China’s urban agglomerations in 2019 was the highest, followed by economic and social resilience. (2) Promoting urban agglomerations had higher resilience scores in the three dimensions, especially in the economic dimension. Growing urban agglomerations had low resilience values on the whole, especially economic resilience. (3) The mean coupling coordination degree of economic-social-ecological resilience ranged from near-incoordination to narrow balance. (4) The coupling coordination degree between the two coincided with the positioning of existing urban agglomerations. (5) Economic resilience had the most significant impact on the coupling coordination. Finally, we give differentiated countermeasures to improve the resilience of urban agglomerations. This study aims to contribute to the promotion of urban resilience research, and helps to plan and design more rational urban economic-social-ecological systems, thereby enhancing the ability of cities to cope with any uncertainties and contingencies.

## 1. Introduction

The term “resilience” was first applied to the field of physics and then gradually appeared in the field of risk governance, with similar concepts such as “susceptibility” and “adaptive capacity”. In 1988, Wildavsky first proposed six essential characteristics of resilient systems: homeostasis, omnivory, high efficiency, flatness, buffering, and redundancy [1]. In recent years, resilience has gradually become the dominant concept in risk governance and is mostly applied at the meso-micro scale, such as community resilience—the ability of a neighborhood or other communities with certain boundaries to cope with and adapt to external changes and interventions, including the ability to dissipate disturbances, self-organize and cope with external pressure [2]. Resilience has also become a hotspot in interdisciplinary research fields such as psychology, sociology, ecology and disaster science.

With the accelerating process of global urbanization since the beginning of the 21st century, land pollution, shortage of resources and energy, traffic congestion and other problems have exerted a negative impact on the ecosystem and sustainable system, and also posed a threat to human living environment [3]. Urban resilience, which was first introduced at the United Nations Sustainable Development Summit in 2002, refers to the ability of urban systems and regions to achieve normal functioning of public safety, social order, and economical construction by reasonably preparing for, buffering, and responding to uncertainty perturbations [4]. Organizations such as the United Nations Office for Disaster Risk Reduction (UNDRR) and the IPCC have since applied the term to sustainable development policies and practices [5]. The 2008 global financial crisis heightened awareness of the importance of urban resilience [6]. The Rockefeller Foundation started the “100 Resilient Cities” project in 2013, a new initiative to focus on urban resilience [7]. The Third United Nations Conference on Housing and Sustainable Urban Development in 2016 included the promotion of “urban ecology and resilience” as one of the core parts of the New Urban Agenda [8]. The outbreak of COVID-19 in late 2019 and the increasingly tense international situation have produced a huge impact on urban systems, making urban resilience once again a concern for governments and residents alike [9].

Urban resilience is currently a new idea and tool for urban risk management that has received widespread attention and recognition, and its study can help all elements of the urban system to organize and coordinate, cope with risks, and adapt to the uncertainty of future development. The studies available have defined urban resilience as a complex system involving economic, social and ecological dimensions and explored specific indicators of urban resilience based on these three dimensions. Ernstson et al. [10] identified a variety of slowly changing variables (e.g., water consumption, land use change, community inequality), technological and social networks (e.g., interactions between energy sources such as water, electricity, oil, and human interactions), urban innovation capacity (e.g., number of patents, number of R&D practitioners, and wealth per capita), and civic engagement as factors influencing urban resilience. Meerow et al. [11] emphasized the importance of electricity, transportation, and green infrastructure for the resilience of cities to climate change. Hudec et al. [12] evaluated urban resilience differences in Slovakia in response to the financial crisis using the Resilience Index (RCI) and 12 indicators in 3 dimensions: economic (e.g., income equality, economic diversification, regional affordability, business environment), sociodemographic (e.g., number of educated, healthy populations, people out of poverty, people insured), and community connectivity (e.g., civic infrastructure, metropolitan stability, housing ownership, voter participation). Cutter et al. [13] conducted a study based on 27 regional resilience evaluation indicators in 5 dimensions: social, economic, infrastructure, institutional, and environmental. Huang Guanying et al. [14] identified 12 critical influencing factors and mechanisms of urban resilience in China based on the 4R framework using the DEMATEL-ISM method. Bai Limin et al. [15] constructed a comprehensive measure index system for urban resilience using 28 indicators from economic, social, ecological, and infrastructure systems. According to the investigation, current studies on urban resilience are generally based on three dimensions of economic resilience, social resilience and ecological resilience, but there is no unified standard for indicators in each dimension. The existing studies mainly focus on the evaluation of resilience analysis and the construction of resilience mechanisms, with no systematic research on the coupling and coordination of the different dimension of resilience. Besides, scholars mainly devote their efforts to the research on the theoretical level, and the existing quantitative studies are mostly based on empirical analysis of cross-sectional data of a certain district or city, with a relative lack of research on the macro-scale.

The concept of urban agglomeration is derived from the theory of metropolitan interlocking region proposed by French economist Gottmann in 1957 [16]. It is a high-end development form that emerges when the regional urbanization reaches a high level, and the highest level of spatial organization in the mature stage of urban development—specifically, a huge, multi-core and multi-level, continuous urban agglomeration formed by a regional central city, or one or two relatively sub-level megacities, as the core [17]. The development of urban agglomerations has become a significant symbol, widely used globally, to measure the level of economic and social development of a country, representing the region’s competitiveness and the level of industrial intensification and chain development [18]. The current urban agglomerations, as organic complexes for coordinated development, can drive the growth of their regions. Thus, they will focus on future development planning for different regions.

This paper is organized into four sections. At the beginning, the interactions between economic resilience, social resilience and ecological resilience are analyzed, followed by a brief insight into the methodology and study area, and finally, the different dimensions of resilience and coupled coordination are assessed and analyzed. The study aims to determine the current urban resilience situation and the synergistic relationship between different dimensions, and provide a comparative reference for the research on urban resilience after the outbreak of the epidemic. The study is designed to gain insights into the ability of urban agglomerations at different levels to cope with risks and uncertainties, in the absence of an emergency context, and to provide a basis for decision-making for their sustainable and high-quality development. This study contributes to the promotion of urban resilience research, and helps plan and design more rational urban economic-social-ecological systems, thereby enhancing the ability of cities to cope with any uncertainties and contingencies.

## 2. Theoretical Model Construction

Adaptive governance theory suggests that by coordinating the relationships between the environment, economy, and society, resilience management strategies can be established, and the state of complex adaptive systems can be regulated to cope with any nonlinear change, uncertainty, and complexity. In this paper, we try to analyze and discuss the interaction of economic-social-ecological resilience, as shown in Figure 1.

Economic resilience (EMR) provides a basic material guarantee for the development of society in many ways. In return, the enhancement of social resilience will improve the conditions of economic factors and contribute to the advance of economic resilience. Economic development is a prerequisite for urbanization and provides material security for the urbanization process, while the benefits of intra-city coordination brought by urbanization boost economic growth [19]. The population age structure and population size affect the economy mainly through three paths—investment, import and export trade, and consumption. In addition, increased social resilience (SR) contributes to the consistent improvement of residents’ living quality, thus driving further economic activities [20]. In line with this, economic development affects regional fertility rates and socio-cultural values, and changes in population size ultimately affect economic resilience [21]. Economic resilience and ecological resilience react and compete against each other. Economic resilience negatively affects ecological resilience in localized areas in the form of the spatial agglomeration of capital and labor [22].With the further growth of the economy, economic resilience provides more material support for the restoration of the living environment to enhance ecological resilience (EGR) when resources and ecological endowments limit a faster pace of economic growth. Therefore, ecological resilience can also affect economic resilience by changing supply efficiency. The ecological environment is interconnected with political culture. Some scholars have proposed a pressure-pulse dynamics (PPD) framework for long-term socio-ecological systems research, holding some conventional ecological changes (e.g., fire, drought) and long-term slow changes (e.g., changes in land use types) act together to alter the structure of biological species and ecological functions. In this way, ecosystem services (including food and carbon cycling) are affected, ultimately altering human actions, which in turn leads to ecological changes [23].

## 3. Study Area and Methods

### 3.1. Study Area Selection

This paper is devoted to the study of 197 prefecture-level cities (excluding autonomous prefectures) in 19 urban agglomerations (“5 + 5 + 9”) proposed in the 14th Five-Year Plan [24], including 5 promoting urban agglomerations (Beijing-Tianjin-Hebei, Yangtze River Delta, Pearl River Delta, Chengdu-Chongqing, and Middle Yangtze), 5 developing urban agglomerations of (Shandong Peninsula, West Taiwan Straits, Central Plains, Guanzhong Plain, and Southern Guangxi), and 9 growing urban agglomerations (Harbin-Changchun, Central and Southern Liaoning, Northern Tianshan Mountains, Hohhot-Baotou-Erdos-Yulin, Lanzhou-Xining, Ningxia-Yellow River, Central Shanxi, Central Guizhou, and Central Yunnan) [25] (Figure 2).

### 3.2. Data Sources and Study Methods

This paper is based on the economic, social and ecological data of Chinese cities in 2019, and the study allows an objective analysis of the development of urban agglomerations as COVID-19 did not appear until the end of 2019. The data were obtained from the China Urban Statistical Yearbook and the statistical data of provinces and cities in 2019. The logical framework of the study is shown in Figure 3.

#### 3.2.1. Fuzzy Logic Method

This paper refines the fuzzy logic reasoning method to measure urban economic resilience (EMR), social resilience (SR) and ecological resilience (EGR). Compared to classical logic methods, fuzzy logic reasoning methods can help capture the approximate and imprecise nature of the real world more effectively, especially when the data are qualitative, inaccurate, or uncertain [26]. Moreover, the fuzzy logic reasoning method is modelled by fuzzy “if-then” rules to help make the measurement results closer to reality [27]. The stability, diversity, innovativeness and resilience of urban economic resilience (EMR), social resilience (SR) and ecological resilience (EGR) are fuzzy concepts with clear connotations but unclear extensions. Therefore, they are suitable for measurement by the fuzzy logic reasoning method.

Firstly, the input parameters are fuzzed. Suppose that there are n input parameters in domains $X_{1},X_{2},\cdots ,X_{n}$ and the output parameter OP is in the domain Y. To capture the uncertainty associated with the collected data, the membership functions of both the input and output parameters are expressed as triangular fuzzy numbers. Next, the positive and negative directions of all indicators for regional economic resilience are analyzed, and each affiliation function is expressed as a vector of the domain X.

Then, the fuzzy rules (if-then rules) are formulated. There are 3^n^ fuzzy rules according to the number of indicators (n), and the same rules are merged using MATLAB with aggregation of the left reasoning relations to construct the fuzzy rule processor. The if-then rule i (i=1,2,⋯,m) is:(1)$$R_{i}{\text{:}\mathrm{if}\;x}_{1}{\;\mathrm{is}\;F}_{1}^{i},x_{2}{\;\mathrm{is}\;F}_{2}^{i},\ldots ,x_{n}{\;\mathrm{is}\;F}_{n}^{i},{\mathrm{then}\;y\;\mathrm{is}\;F}_{0}^{i}$$
where $x_{j}\in X_{j},y\in Y(j=1,2,\cdots ,n)$ and $F_{j}^{i}(j=1,2,\cdots ,n)$ are triangular fuzzy numbers and represent the qualitative descriptions of the corresponding input indexes, respectively, and $F_{0}^{i}$ is also a triangular fuzzy number which represents the qualitative description of the corresponding output index.

The intensity $\beta_{i}$ of rule i is calculated by the fuzzy intersection operation as follows:(2)$$\begin{array}{l} \beta_{i}=\max\{\min\{\mu_{F_{1}^{i}}(x),\mu_{G_{1}}(x)\}:x\in X_{1}\}\wedge \max\{\min\{\mu_{F_{2}^{i}}(x),\mu_{G_{2}}(x)\}:x\in X_{2}\}\\ \wedge \cdots \wedge \max\{\min\{\mu_{F_{n}^{i}}(x),\mu_{G_{n}}(x)\}:x\in X_{n}\} \end{array}$$
where, $\mu_{F_{j}^{i}}(j=1,2,\cdots ,n)$ is the membership function that qualitatively describes the corresponding fuzzy set of $F_{j}^{i}(j=1,2,\cdots ,n)$ in rule $R_{i}$, and $\mu_{G_{j}}(j=1,2,\cdots ,n)$ is the membership function of fuzzy input given in the form of triangular fuzzy numbers.

Then, the fuzzy output of each rule is calculated by fuzzy intersection operation.
(3)$$\nu_{F_{j}^{i}}(y)=\min\{\beta_{i},\mu_{F_{0}^{i}}(y)\}$$
where, $\beta_{i}$ is the strength of $R_{i}$ in rule i; $\mu_{F_{0}^{i}}$ is the membership function of the qualitative description $F_{0}^{i}$ of the output index, y∈Y. Then the rule i is calculated and aggregated by the fuzzy method.
(4)$$\nu_{F_{0}}(y)=\max\{\nu_{F_{0}^{1}}(y),\nu_{F_{0}^{2}}(y),\cdots ,\nu_{F_{0}^{m}}(y)\}$$
where, $\nu_{F_{0}^{i}}$ is the membership function of fuzzy output derived from $R_{i}$ in rule i; and n is the total number of rules, y∈Y.

In the end, the output obtained from aggregation is defuzzified using the area prime method to get the final resilience results output as follows:(5)$$OP=\frac{\sum_{l=1}^{k}y_{l}\cdot \mu_{F_{0}}(y_{l})}{\sum_{l=1}^{k}\mu_{F_{0}}(y_{l})}$$
where, $y_{j}\in Y$ is Y quantized by k.

#### 3.2.2. Coupling Coordination Degree Model of Three Dimensions

Coupling is a phenomenon in which two or more systems interact with each other by acting upon each other to the extent that they form a whole and act together [28]. Based on the obtained resilience evaluation results, this paper measures the relationship between the resilience dimensions using the coupling coordination degree model. The coupling degree model of urban economic resilience (EMR), social resilience (SR) and ecological resilience (EGR) is calculated [29] as follows:(6)$$C={\left\{\frac{f(emr)\times g(sr)\times h(egr)}{{\left[\frac{f(emr)+g(sr)+h(egr)}{3}\right]}^{3}}\right\}}^{\frac{1}{3}}$$
where, *C* represents the coupling degree with a value in the range of 0–1; f(emr), g(sr) and h(egr) are the scores of EMR, SR and EGR, respectively. 

The coupling degree reflects the strength of the interaction between EMR, SR and EGR. Still, it is hard to measure how well they develop in a coordinated manner. To this end, the coupling coordination degree model of the three is further constructed as follows:(7)$$D=\sqrt{C\times T},T=\alpha f(emr)+\beta f(sr)+\chi h(egr)$$
where, *D* represents the coupling coordination degree, *C* represents the coupling degree, and *T* is the comprehensive evaluation coefficient of the three subsystems; α, β and χ are the weights to be determined and according to the relevant study, their values are 0.4, 0.3, and 0.3, respectively.

#### 3.2.3. Coupling Coordination Degree Model of Two Dimensions

A coupling coordination degree model between two is established to further explore the interconnection and synergy between different dimensions of resilience.
(8)$$C=\sqrt{\frac{A\times B}{{\left(\frac{A+B}{2}\right)}^{2}}}$$
(9)T=αA+βB
(10)$$D=\sqrt{C\times T}$$

*C* represents the coupling degree of *A* and *B* with a value in the range of [0, 1]; *T* is the combined development index of *A* and *B*; α and β are coefficients to be determined; *D* represents the coupling coordination degree.

### 3.3. Index Selection

Urban resilience is affected by the joint coordination of economic-social-ecological dimensions and requires that these three systems be considered as a whole. The selection of indexes also requires consideration of the interactions between systems. Therefore, the indexes were selected with reference to cutting-edge urban resilience research [15,30,31,32,33,34].

For economic resilience, given that economic development is closely related to factor endowment, the main proxies considered here are concentration of human capital and infrastructure development. Consequently, we chose GDP per capita and investment in urban facilities as variable indicators. Past research has also demonstrated that regional competitiveness, innovation, and agglomeration economies are also major determinants of a region’s ability to withstand or respond to a crisis [35]. Therefore, the number of enterprises above designated size, and proportion of total import and export, are also included in the index. The internal workings of the social resilience system are complex, and involve the impact of individuals’ psychology as well as decisions in society. In the current version of our model, for several reasons, we did not focus on these individual-level factors. Instead, we assessed it from a more objective perspective—the dynamics of society as a whole, and the degree of individual job stability. Therefore, we chose indexes such as the aging rate, the total population with tertiary education and above, number of employees in the tertiary industry, number of colleges and universities, R&D practitioners, and number of registered unemployed persons in urban areas. For ecological resilience, we chose the green coverage of built-up areas and per capita water resources representing ecological resources, and take industrial smoke (dust) emissions, centralized treatment rate of sewage treatment plants, harmless disposal rate of domestic waste, comprehensive utilization rate of general industrial solid waste, proportion of construction land area and AQI (Air Quality Index) to represent environmental protection. The weights of the indexes are determined by the fuzzy logic method. All indexes are listed in Table 1.

## 4. Economic-Social-Ecological Resilience

### 4.1. Economic-Social-Ecological Resilience Measurement Results

China’s urban agglomerations had the highest mean ecological resilience in 2019, followed by economic and social resilience. The ecological resilience was scored in the range of 0.33–0.67, with the economic resilience in the range of 0.20–0.69, and social resilience in the range of 0.2–0.4.

Economic resilience was generally higher in promoting and developing urban agglomerations than in growing urban agglomerations. The resilience values of promoting urban agglomerations in the three dimensions were higher, especially the economic resilience. Economic resilience and social resilience had the same mean for developing urban agglomerations, but the economic resilience was at a low level. The overall resilience value of growing urban agglomerations was low, especially the economic resilience (Figure 4).

#### 4.1.1. Economic Resilience

The economic resilience score was consistent with the positioning of existing urban agglomerations, that is, optimizing and developing urban agglomerations generally had higher values than growing urban agglomerations.

(1) The mean economic resilience values of promoting urban agglomerations remained at 0.31–0.46. Apart from the Pearl River Delta and Yangtze River Delta urban agglomerations, all exhibited a “center-periphery” spatial development pattern, i.e., only some regions were extraordinarily resilient and raised the overall average. Only Beijing, Tianjin and Shijiazhuang performed well in the Beijing-Tianjin-Hebei urban agglomeration, with other regions scored below 0.35 and some even below 0.3 in resilience. In the Chengdu-Chongqing urban agglomeration, Chengdu and Chongqing showed higher economic resilience, while other regions were at an average level. However, uneven development was also found in the urban agglomeration in Middle Yangtze, there were more regions with higher economic resilience, and the prefecture-level cities at a low level all remained above 0.31, with comparatively even development. The Pearl River Delta and Yangtze River Delta urban agglomerations developed evenly and satisfactorily, and both had an average economic resilience value of 0.46.

(2) The developing urban agglomerations all showed an unbalanced development pattern in parts with different degrees of imbalance, except Southern Guangxi urban agglomeration, whose economic resilience remained at a low level of balance. Shandong Peninsula urban agglomeration had the highest overall resilience value, and Jinan and Qingdao acted as its growth poles with economic resilience values of 0.58 and 0.61 respectively. In addition, there was a mild degree of imbalance in the Shandong Peninsula urban agglomeration, and other cities such as Yantai and Weifang scored at or above 0.4. Guanzhong urban agglomeration had the most serious imbalance, and the economic resilience of Xi’an, the city at the highest level, reached up to 0.5 or above, compared to only 0.21–0.29 in other regions.

(3) The mean economic resilience of growing urban agglomerations was below 0.3, with the lowest level in Harbin-Changchun and Lanzhou-Xining urban agglomerations, which were 0.26 and 0.27, respectively. There were no significant imbalances within the urban agglomerations. The results for the urban agglomerations in Central Yunnan and Central Guizhou were slightly higher than the real level due to the small number of cities included and a shortage of data for autonomous prefectures. The urban agglomerations with low economic resilience were roughly located in northwest and northeast China, mostly growing urban agglomerations.

#### 4.1.2. Social Resilience

Social resilience fluctuated in the range of 0.2–0.4, with a few rising to higher than 0.6 and no regions below 0.2.

(1) In general, the mean social resilience of promoting urban agglomerations was above 0.35. However, the urban agglomeration in Middle Yangtze urban agglomeration was a special case with a mean value of only 0.32, and it was also one of the few urban agglomerations with lower social resilience than economic resilience. In addition, there were obvious differences in the resilience values within each urban agglomeration, with 1–3 areas with very high social resilience of 0.5 or more within each urban agglomeration, while the resilience values of other prefecture-level cities were all largely similar, at around 0.3. For example, Chengdu and Chongqing in Chengdu-Chongqing urban agglomeration, Beijing in Beijing-Tianjin-Hebei urban agglomeration, Wuhan in the urban agglomeration in Middle Yangtze urban agglomeration, Shanghai and Nanjing in Yangtze River Delta urban agglomeration, and Shenzhen in Pearl River Delta urban agglomeration had a significantly higher resilience value than other cities in their urban agglomerations. However, Chongqing, a city in the Chengdu-Chongqing urban agglomeration, despite a high aging rate and a gap with other central cities in population development, had better social infrastructure security and social environment, with a larger population obtained higher education and a larger road area, as well as a lower level of unemployment social resilience, all which contributed to its high social resilience. Although regional imbalances were found within the Yangtze River Delta and Pearl River Delta urban agglomerations, there were many prefecture-level cities with a resilience value of around 0.4, indicating a little better state than other promoting urban agglomerations on the whole.

(2) Some regions in emerging urban agglomerations were also found to drive up the mean value of the overall urban agglomerations. Nanning, in the Southern Guangxi urban agglomeration, had the greatest social resilience of 0.38, while Beihai came in second with 0.32. Xiamen has the greatest social resilience of 0.42 among the cities in the West Taiwan Straits urban agglomeration, followed by Wenzhou with 0.37. In contrast to the promoting urban agglomeration, the growing urban agglomeration had no cities with a cliff gap greater than 0.1.

(3) Among the growing urban agglomerations, Lanzhou-Xining urban agglomeration and the urban agglomeration in Ningxia-Yellow River urban agglomeration had the lowest mean values, both being 0.29. It was comparatively even within urban agglomerations, with slight disparity in social resilience between cities.

#### 4.1.3. Ecological Resilience

With the highest ecological resilience score, the ecological resilience of urban agglomerations remained proportional to their positioning, except for some urban agglomerations.

(1) The mean value of ecological resilience promoting urban agglomerations was above 0.48, but there were inter-regional differences in resilience within urban agglomerations. For example, the ecological resilience of Xingtai in Beijing-Tianjin-Hebei urban agglomeration was scored 0.66. The ecological resilience values of Huanggang and Ji’an in the urban agglomeration in Middle Yangtze River were 0.66 and 0.64, respectively, while not more than 0.6 for the rest regions. The exceptions were Yangtze River Delta and Pearl River Delta urban agglomerations, where the ecological resilience values of the regions were at a high level of equilibrium, basically around 0.55. It should be noted that the economic resilience of promoting urban agglomerations was much higher than that of the cities in the surrounding areas, and their social resilience was also comparatively high, except for their average ecological resilience. The results showed that these cities performed well in terms of financial development and education, attracting a large workforce and leading to a spread of infrastructure, but with a corresponding degree of ecological burden.

(2) The mean value of ecological resilience of developing urban agglomerations was 0.51, and only Guanzhong’s urban agglomeration was at a low level, with a mean value of only 0.48. In addition, apart from the Shandong Peninsula urban agglomeration, there were also urban agglomeration containing some regions scored far higher than those in their surrounding areas. For example, the ecological resilience of Fuzhou in West Taiwan Straits urban agglomeration reached 0.59, and that of Zhengzhou in Central Plains urban agglomeration was 0.63. Unlike the pattern presented by promoting urban agglomerations, the economic resilience and social resilience of the above regions were higher, showing a better synergy between the economic-social-ecological subsystems.

(3) The mean value of ecological resilience of growing urban agglomerations ranged from 0.39 to 0.50. Hohhot-Baotou-Erdos-Yulin urban agglomeration had the lowest ecological resilience with a mean value of 0.39, while the urban agglomerations in the Northern Tianshan Mountains and in Central and Southern Liaoning urban agglomeration had an ecological resilience of 0.50. Compared to promoting and developing urban agglomerations, growing urban agglomerations were generally less ecologically resilient but more internally balanced. The regions with the lowest ecological resilience were distributed in the northwest and northeast China, such as Baiyin in Lanzhou-Xining urban agglomeration and Wuzhong in the urban agglomeration in Ningxia-Yellow River, showing that the environmental problems were still prominent due to the fragile ecological environment in the northwest.

### 4.2. The Coupling Coordination Degree of Economic, Social and Ecological Resilience

The coupling coordination degree indicates the strength of interactions between different dimensions of resilience in a region and the level of coordinated development. By measuring the coupling coordination degree of national urban agglomerations, we find that higher economic resilience and social resilience usually lead to a higher coupling coordination degree. However, the coupling coordination degree is also constrained by ecological resilience. The coupling coordination degree is divided into 5 levels as follows: imbalance (0.2 < D ≤0.4), moderate imbalance (0.4 < D ≤ 0.6), low coordination (0.6 < D ≤ 0.7), moderate coordination (0.7 < D ≤ 0.8), and advanced coordination (0.8 < D ≤ 1) [36].

The coupling coordination degree of economic-social-ecological resilience of China’s urban agglomerations is proportional to their positioning, and the mean coupling coordination degree of economic-social-ecological resilience ranges from moderate imbalance to low coordination. Regarding spatial distribution, areas with advanced coupling coordination were mainly distributed in promoting urban agglomerations. Growing urban agglomerations have the lowest mean value of coupling coordination which coincides with the findings of another study [20], see Figure 5.

(1) The coupling coordination degree of economic-social-ecological resilience of promoting urban agglomerations was at a low coordination level. Yangtze River Delta and Pearl River Delta urban agglomerations had the highest mean values of coupling coordination, at 0.66 and 0.68, respectively. There were no significant disparities among its internal regions. Some cities in the urban agglomeration in the Middle Yangtze River, as well as Chengdu-Chongqing and Beijing-Tianjin-Hebei urban agglomerations had a significantly higher coupling coordination degree than other regions, such as Chongqing, Beijing and Tianjin, which even reached the level of moderate coordination. These higher scoring regions were also doing well in economies with higher economic resilience.

(2) The mean coupling coordination degree of economic-social-ecological resilience of developing urban agglomerations was at a low coordination level. Furthermore, there were regional imbalances in urban agglomerations, and some cities with advanced coupling coordination were comparable to cities promoting urban agglomerations. For example, Qingdao in Shandong Peninsula urban agglomeration was in the moderate coordination range, much higher than the cities promoting urban agglomerations such as Taizhou, Shaoxing, Zhuhai, and Changsha. Guanzhong urban agglomeration was the only one with a mean value below 0.6 among the developing urban agglomerations, which was at a level of moderate imbalance.

(3) The coupling coordination of economic-social-ecological resilience of growing urban agglomerations mainly was at a level of moderate imbalance. Their coupling coordination degree was in the range of 0.51–0.60. However, there were probably some cities with better performance, such as Guiyang in Central Guizhou urban agglomeration, Dalian and Shenyang in Central and Southern Liaoning urban agglomeration, Kunming in Central Yunnan urban agglomeration, Urumqi in Northern Tianshan Mountains urban agglomeration, Taiyuan in Central Shanxi urban agglomeration, which was at a level of low coordination.

### 4.3. The Coupling Coordination Degree of Economic-Social-Ecological Resilience of Two

The coupling coordination of the three reflects the extent of synergy of economic-social-ecological resilience in urban agglomerations, but does not describe the relationship between the dimensions of synergy. Therefore, the coupling coordination degree model of two is used to analyze the synergy relationship among the dimensions of economic-social-ecological resilience in urban agglomerations, see Figure 6.

#### 4.3.1. Coupling Coordination Degree of Economic-Social Resilience

The coupling coordination degree of economic and social resilience in China’s urban agglomerations were in the range of 0.48–0.79, a moderate imbalance to low coordination level. Most of the regions at the level of “Moderate imbalance” and “Low coordination” were in promoting and developing urban agglomerations, while the regions with “Moderate imbalance” were mainly in growing urban agglomerations. In addition, there were great spatial differences in the degree of coupling coordination within the urban agglomerations.

(1) The mean value of the coupling coordination degree of economic and social resilience of promoting urban agglomerations was in the range of 0.56–0.65, and it was at the level of moderate imbalance to low coordination on the whole for all urban agglomerations. Some regions were at a moderate coordination level, higher than the mean value of urban agglomerations. For example, Beijing in Beijing-Tianjin-Hebei urban agglomeration and Wuhan in the urban agglomeration in the Middle Yangtze River both scored high, significantly above the surrounding areas, and showing spatial imbalance. Of all promoting urban agglomerations, Chengdu-Chongqing urban agglomeration was at the lowest level, even lagging behind some cities in the developing urban agglomerations, with a mean value of 0.58. The coupling coordination degree of economic-social resilience in Ya’an, Guang’an and Dazhou, the cities in Chengdu-Chongqing urban agglomeration, was about 0.5, lower than that in most regions of China.

(2) The mean coupling coordination degree of economic-social resilience in the developing urban agglomerations was in the range of 0.53–0.60, a level of low coordination. Jinan and Qingdao in Shandong Peninsula urban agglomeration, Xiamen in West Taiwan Straits urban agglomeration and Xi’an in Guanzhong urban agglomeration had an economic-social coupling coordination degree of over 0.7, a moderate coordination level. There was a small gap between Southern Guangxi and Central Plains urban agglomerations.

(3) The coupling coordination degree of economic-social resilience in growing urban agglomerations was in the range of 0.52–0.57, a level of moderate imbalance. Harbin-Changchun and Lanzhou-Xining urban agglomerations had the lowest values, which were 0.55 and 0.53, respectively. And a “core-periphery structure” was exhibited in the urban agglomerations, and the regions with noticeable performance, such as Guiyang in Guiyang in Central Guizhou urban agglomeration, Shenyang and Dalian in Central and Southern Liaoning urban agglomeration, Harbin in Harbin-Changchun urban agglomeration, Urumqi in the urban agglomeration in Northern Tianshan Mountains, and Hohhot in Hohhot-Baotou-Erdos-Yulin urban agglomeration, whose values were even higher than those of the regions in promoting urban agglomerations.

#### 4.3.2. Coupling Coordination Degree of Social-Ecological Resilience

The coupling coordination degree of social-ecological resilience of the 19 urban agglomerations ranged from 0.53 to 0.78, a level of low coordination or moderate coordination. Promoting and developing urban agglomerations had similar mean coupling coordination degrees of social-ecological resilience, while the growing urban agglomerations had a lower mean value. The coupling coordination degree between different city cluster types was less disparate, and its spatial performance was more balanced than the coupling coordination degree of economic-social resilience, which was also reflected in all urban agglomerations.

(1) The coupling coordination degree of social-ecological resilience in the promoting urban agglomeration was in the range of 0.64–0.68, a low coordination level. Some regions in urban agglomerations achieved a higher level of coordination, such as Chongqing in Chengdu-Chongqing urban agglomeration and Shenzhen in Pearl River Delta urban agglomeration, which was above 0.7, a moderate coordination level. Despite the high social resilience and ecological resilience in promoting urban agglomerations, because the distribution areas with good social resilience performance were not the same as those with good ecological resilience, and the two interacted with each other to produce average coupling coordination of social-ecological resilience in most areas, they were not significantly higher than those of developing and growing urban agglomerations.

(2) The coupling coordination degree of social-ecological resilience in the developing urban agglomeration was in the range of 0.60–0.65, also a low coordination level. The coupling coordination degree of social-ecological resilience in Guanzhong urban agglomeration was the lowest, more than 0.3, lower than that of other urban agglomerations. Developing urban agglomerations performed better overall, indicating a good level of coordinated development between social resilience and ecological resilience. The education and infrastructure improvement went hand in hand with reducing pollution emissions, showing that local governments had some success in social management and social security.

(3) The value was in the range of 0.59–0.63 for the growing urban agglomerations, a level of moderate imbalance to low coordination, with the lowest mean values in the urban agglomeration in Ningxia-Yellow River and Lanzhou-Xining urban agglomeration in northwest China. It indicates that the fragile ecological environment in the northwest region negatively impactssocial infrastructure development and population density, and the two systems are less able to adapt and coordinate with each other.

#### 4.3.3. Coupling Coordination Degree of Economic-Ecological Resilience

The coupling coordination degree of economic-ecological resilience was higher than economic-social resilience but lower than social-ecological resilience. In general, the coupling coordination of economic-ecological resilience ranged from 0.51 to 0.80, a level of low coordination to advanced coordination. The coupling coordination degree of economic-ecological resilience was generally low in north China, especially in the northwest and northeast, that is, growing urban agglomerations were scored low.

(1) The mean coupling coordination degree of economic-ecological resilience in the promoting urban agglomeration was in the range of 0.60–0.70, a low coordination level. The highest mean value of coupling coordination was found in Pearl River Delta urban agglomeration, and all other regions were scored 0.68 and above in the urban agglomeration, except for Zhaoqing with a value of 0.64. The Chengdu-Chongqing urban agglomeration had the lowest mean value, even lower than that of the Northern Tianshan Mountains urban agglomeration, one of growing urban agglomerations, and it was also an urban agglomerations with the largest spatial difference. The coupling coordination degree of Dazhou and Guang’an in Chengdu-Chongqing urban agglomeration was only about 0.3, but Chengdu reached 0.80, an advanced coordination level.

(2) The mean coupling coordination degree of economic-ecological resilience in the developing urban agglomeration was in the range of 0.60–0.65, also a low coordination level. Guanzhong urban agglomeration had the lowest value, and the coupling coordination degree of the covered cities Tianshui and Shangluo was about 0.52, mainly due to the severe spatial imbalance of economic resilience and low ecological resilience in the Guanzhong urban agglomeration.

(3) The mean coupling coordination degree of economic-ecological resilience ranged from 0.56 to 0.61, a level of moderate imbalance to low coordination, lower than that of promoting and developing urban agglomerations. Except for the urban agglomerations in Central Yunnan and in the Northern Tianshan Mountains urban agglomeration, all other urban agglomerations were at a level of low coordination. The lowest mean values of coupling coordination were found in Lanzhou-Xining, Harbin-Changchun, and Ningxia-Yellow River urban agglomerations.

### 4.4. The Influence of Different Dimensions on Coupling Coordination

There are mutual influences among urban agglomerations. This paper establishes an individual fixed effect model, and excludes the individual disturbance terms using cluster-robust standard error. Under the assumption that the resilience among different urban clusters is not correlated, the paper explores the relationship between economic resilience, social resilience, and ecological resilience and the degree of coupling coordination among the three. The specific model is set as follows:$$Y_{i}=\beta_{0}+\beta_{1}X_{1i}+\beta_{2}X_{2i}+\beta_{3}X_{3i}+\mu_{i}+\sigma_{i}$$
where, $Y_{i}$ represents the coupling coordination degree of urban agglomeration i; *X*_1*i*_, *X*_2*i*_ and $X_{3i}$ respectively represent economic resilience, social resilience and ecological resilience of urban agglomeration i; $\mu_{i}$ is used to control the individual fixed effects of each urban agglomeration, and $\sigma_{i}$ is the residual term. Theoretically, a higher level of fixed effects clustering makes more stringent assumptions on the perturbation term, so the lowest level of fixed effects may yield the most robust results, and the fixed-effects model regression results are shown in Table 2.

Table 2 shows the regression results of 19 urban agglomerations, and all three dimensions have independent effects on coupling coordination. It’s clear that the coefficients of economic resilience, social resilience, and ecological resilience were all positive. The *p*-values of all urban agglomerations were less than 0.01 and significant at the 1% level, except for the Hohhot-Baotou-Erdos-Yulin urban agglomeration, the urban agglomeration in the Northern Tianshan Mountains urban agglomeration, the urban agglomeration in Ningxia-Yellow River, and urban agglomerations in Central Yunnan and Central Guizhou with smaller sample sizes, indicating that the joint significance of economic resilience, social resilience, and ecological resilience for their total coupling coordination was excellent. According to the results, economic resilience has the greatest impact on coupling coordination, indicating that the improvement of regional economic resilience was more helpful for coordinating urban economic, social and ecological systems. The ecological resilience of the developing and growing urban agglomerations had a less positive impact on the coupling coordination degree, while the positive effect of resilience of the promoting urban agglomerations was comparatively balanced, indicating that the three were better coordinated.

For the promoting urban agglomerations, the economic resilience, social resilience and ecological resilience in the coastal region and the middle reaches of the Yangtze River all showed a significant positive effect on the coupling coordination degree of economic-social-ecological resilience, indicating that the coupling coordination degree varies with the change of either economic, social and ecological resilience. This is to say that, generally, these urban agglomerations were well developed, and the three were able to achieve good coordination and co-progress. As the economy has developed to a certain stage, these areas have been effective in providing social security and ecological civilization, with economic development, social construction and the ecological environment enhanced simultaneously to a certain extent. The social resilience and ecological resilience of Beijing-Tianjin-Hebei and Chengdu-Chongqing urban agglomerations had an average impact on the coupling coordination degree, indicating that the spatial imbalance in these regions was serious. Currently, it is mainly the economic resilience that drives the overall coupling coordination degree of the system, and it is still necessary to further exert economic advantages with attention to coordination and co-progress between the systems.

For the developing urban agglomerations, the social resilience of Central Plains, Guanzhong and West Taiwan Straits urban agglomerations contributed significantly to the coupling coordination. According to the results of social resilience, the spatial imbalance of social resilience in these urban agglomerations was at a low level, indicating that the continuous improvement of the social environment, social infrastructure, and social system had a significant positive impact on the economic-social-ecological coupling coordination degree.

In the growing urban agglomerations, it was mainly the economic resilience that drove the coupling coordination degree of economic-social-ecological systems, and the social resilience and ecological resilience had less impact, which was related to the current development status of growing urban agglomerations. Currently, there is still a large development space for the urban economy, social security, and other areas that do not move in lockstep with the growth of economic development, and the spatial imbalance is severe.

## 5. Conclusions and Countermeasures

### 5.1. Conclusions

(1) The mean ecological resilience of China’s urban agglomerations in 2019 was the highest, followed by economic and social resilience. The ecological resilience was scored in the range of 0.33–0.67, with the economic resilience in the range of 0.20–0.69, and social resilience in the range of 0.2–0.4. The resilience values of promoting urban agglomerations in the three dimensions were higher, especially the economic resilience. Developing urban agglomerations had the same mean economic resilience and social resilience, but their economic resilience was lower. The overall resilience value of growing urban agglomerations was low, especially the economic resilience.

(2) The mean coupling coordination degree of economic-social-ecological resilience ranged from moderate imbalance to low coordination. The areas with a high coupling coordination degree were mainly distributed in promoting urban agglomerations, while growing urban agglomerations were near incoordination.

(3) The coupling coordination degree of economic-ecological resilience was higher than that of economic-social resilience, but lower than that of social-ecological resilience. The coupling coordination degree of economic-social resilience was in the range of 0.48–0.79, a moderate imbalance or moderate coordination level. The coupling coordination degree of social-ecological resilience was in the range of 0.53–0.78, a level of low coordination to moderate coordination. The coupling coordination degree of economic-ecological resilience was in the range of 0.51–0.80, a level of low coordination to advanced coordination. The coupling coordination degree of the two coincided with the positioning of existing urban agglomerations, i.e., the degree of coupling coordination between the two dimensions of promoting and developing urban agglomerations was generally higher than that of growing urban agglomerations.

(4) Economic resilience had the most significant impact on the coupling coordination degree, i.e., increased regional economic resilience better facilitated coordination between urban economic-social-ecological systems.

### 5.2. Countermeasures

Based on the above findings and the current urban cluster development in China, differentiated strategies for improving the resilience of urban agglomerations are proposed:

(1) Promoting urban agglomerations should focus on spatial balance, improved social security, and strengthen trickle-down functions.

Promoting urban agglomerations, in general, had high resilience values, but at an average level of coordination, and all others were found to have obvious spatial imbalance except for Yangtze River Delta and Pearl River Delta urban agglomerations. Central cities should gradually implement the internal dispersion and external decentralization of urban functions [37], build multiple-core areas and develop sub-center cities to create new development opportunities while driving the development of neighboring cities. Although the economic resilience of promoting urban agglomerations raised the coupling coordination degree, their lower social resilience decreased it. This indicates there is no coordination between economic construction, the level of infrastructure and the organizational capacity of urban management, and further enhancement of construction is still required. The government should strengthen investment in a targeted manner to improve the social security system, improve the residents’ living standards and ensure the sustainable development of the social environment. In addition, promoting urban agglomerations are mostly near the river and the sea in spatial distribution, which gives them natural advantages in logistics, and transportation and information contact. Therefore, they should give full play to the linkage effect and leading effect of high-quality development to carry out spatial reorganization of resources, so as to avoid the negative effects brought by high concentration.

(2) Developing urban agglomerations should continue to accelerate urbanization, upgrade infrastructure, and enhance synergistic development.

The level of economic resilience varies widely among cities in developing urban agglomerations, while the central city resilience value has a certain gap compared to that of promoting urban agglomerations, but the coupling coordination is at a high level. The main problem of developing urban agglomerations is the economic imbalance between regions. To avoid over-concentration of factors, developing urban agglomerations can transfer some industries from economic centers to the peripheral zones of economically backward urban agglomerations [38], with an attempt to establish economic synergy and cooperation between regions, and thus realize regional integration. In addition, developing urban agglomerations with better social resilience, especially the central cities, should take advantage of their better infrastructures and more substantial attractiveness to attract talent, while emphasizing personnel training to improve the quality of the labor, to further enhance urban competitiveness.

(3) Growing urban agglomerations should adjust their industrial structure, enhance economic vitality, and promote high-quality development.

Growing urban agglomerations are located in the remote areas of northwest, northeast and southwest China, and the cities generally have lower economic and social resilience levels than developing and promoting urban agglomerations. Firstly, growing urban agglomerations can only improve their overall economic development by boosting their local characteristics and carrying out industrial restructuring simultaneously. Their ecological resilience and economic-ecological coupling coordination are both low, indicating that the industries have caused some ecological impact and have not been well transitioned into the local economy. It requires that local governments should change their ideas, develop distinctive and green industries, and try to boost secondary and tertiary industries. Secondly, growing urban agglomerations should strengthen ties with promoting urban agglomerations, and raise the spatial allocation efficiency of financial resources within urban agglomerations, to seek high-quality development.

## Figures and Tables

**Figure 1 ijerph-20-00413-f001:**
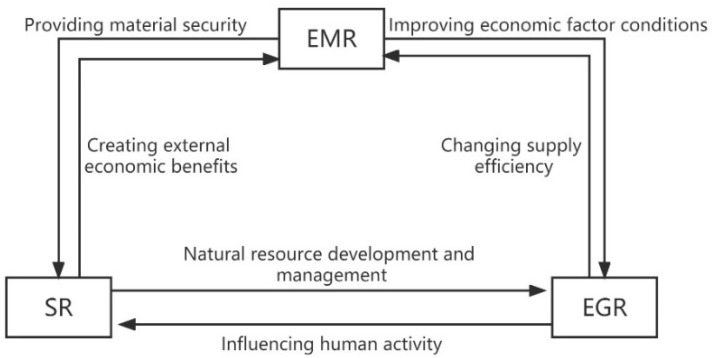
The relationship of economic, social, and ecological resilience.

**Figure 2 ijerph-20-00413-f002:**
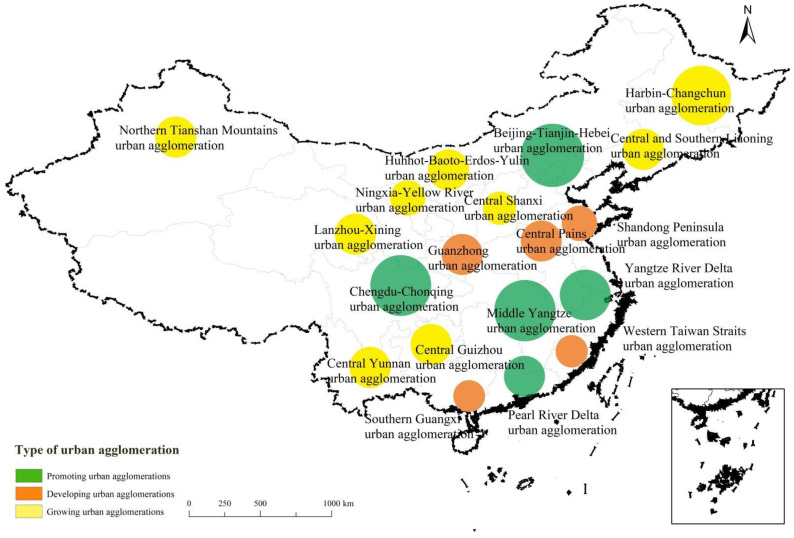
Distribution of China’s urban agglomerations. Note: The map of China is drawn to the standard scale, and the review number is GS(2019)1822. There is no modification to the base.

**Figure 3 ijerph-20-00413-f003:**
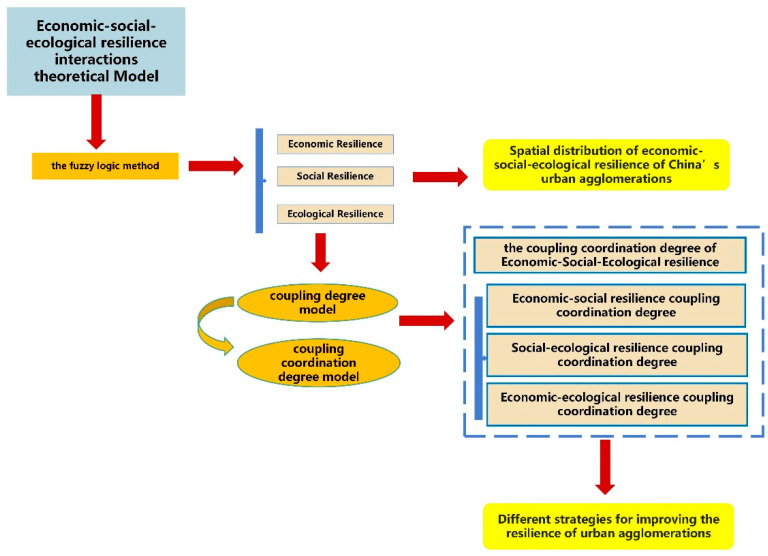
Logical framework of the study.

**Figure 4 ijerph-20-00413-f004:**
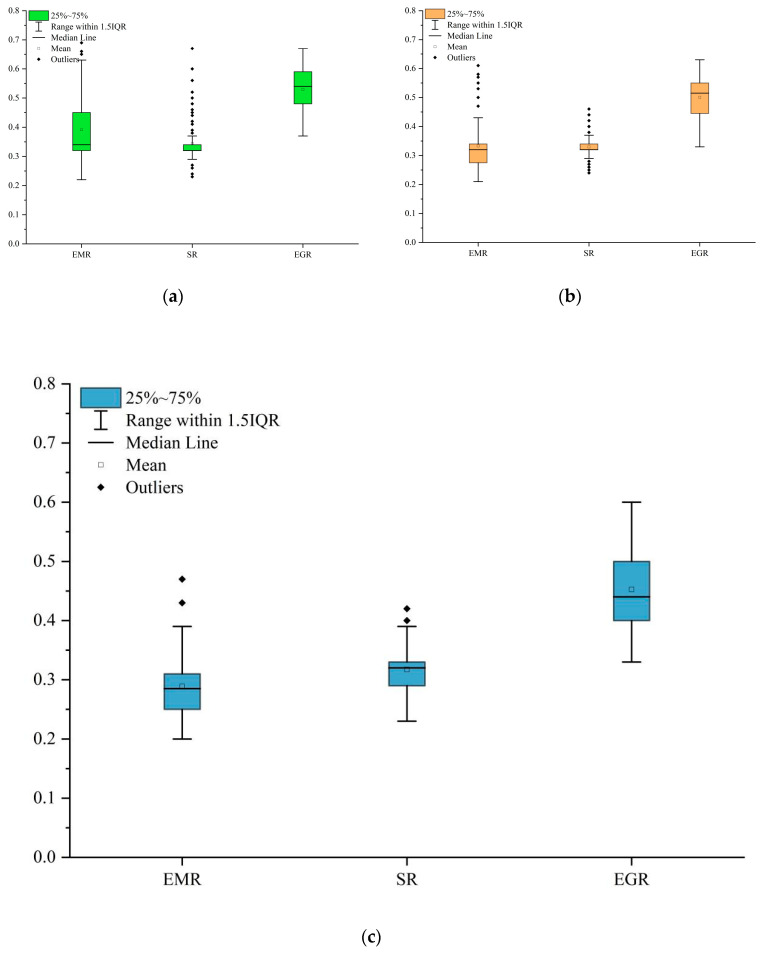
Economic-social-ecological resilience of China’s urban agglomerations. (**a**) Growing urban agglomerations. (**b**) Developing urban agglomerations. (**c**) Promoting urban agglomerations.

**Figure 5 ijerph-20-00413-f005:**
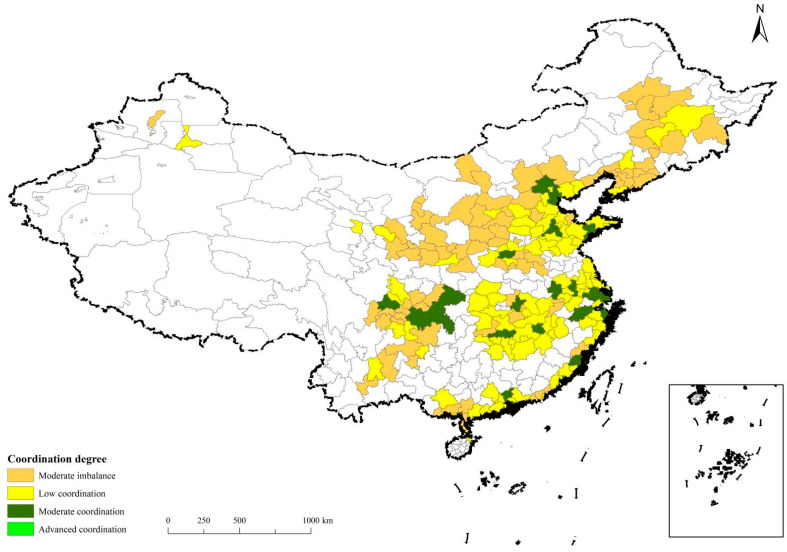
Coupling result of economic-social-ecological resilience in Chinese urban agglomerations.

**Figure 6 ijerph-20-00413-f006:**
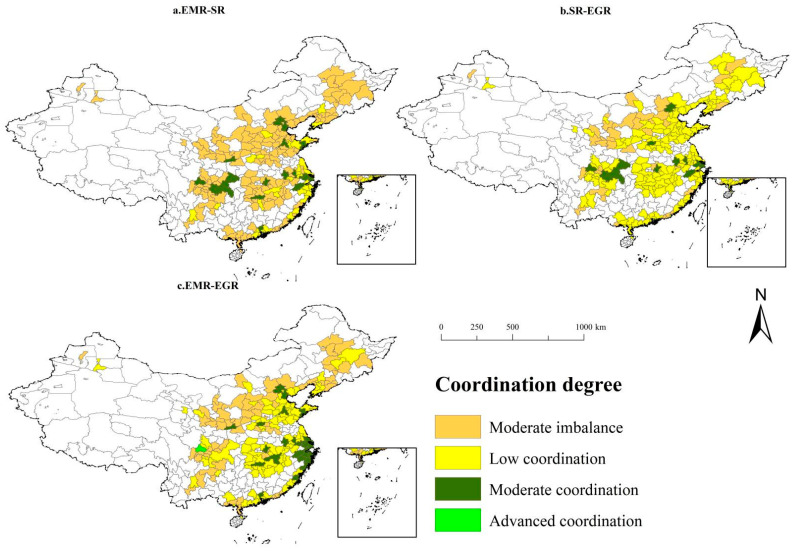
The two coupling coordination degrees.

**Table 1 ijerph-20-00413-t001:** Evaluation index system.

Economic Resilience	Social Resilience	Ecological Resilience
Per capita GDP (10^4^ Yuan)	Actual road area at the end of the year (m^2^)	Green coverage of built-up areas (%)
Proportion of tertiary industry (%)	Total population with tertiary education and above	Per capita water resources (10^4^ m^2^)
Expenditure on science and technology (%)	Number of colleges and universities	Industrial oxynitride emissions (tons)
Expenditure on education (%)	Population density (/km^2^)	Industrial smoke (dust) emissions (tons)
Number of enterprises above designated size	Natural growth rate (%)	Centralized treatment rate of sewage treatment plants (%)
Proportion of total import and export (%)	Aging rate (%)	Harmless disposal rate of domestic waste (%)
Average wage of on-the-job workers (yuan)	Number of employees in the tertiary industry	Comprehensive utilization rate of general industrial solid waste (%)
Investment in urban public facilities construction (10^4^ yuan)	R&D practitioners	Proportion of construction land area (%)
Residents’ savings deposit balance (10^4^ yuan)	Number of registered unemployed persons in urban areas	Air Quality Index

**Table 2 ijerph-20-00413-t002:** Fixed-effects model regression results.

Urban Agglomeration	Coupling Result		EMR	SR	EGR
Central Pains urban agglomeration	EMR-SR-EGR	coefficient	0.198 ***	0.147 ***	0.104 ***
		S.E.	(0.00641)	(0.0159)	(0.00540)
Beijing-Tianjin-Hebei urban agglomeration	EMR-SR-EGR	coefficient	0.199 ***	0.0941 ***	0.107 ***
		S.E.	(0.0173)	(0.0249)	(0.00847)
Lanzhou-Xining urban agglomeration	EMR-SR-EGR	coefficient	0.468 **	--	0.0726 **
		S.E.	(0.00773)	--	(0.00333)
Guanzhong urban agglomeration	EMR-SR-EGR	coefficient	0.192 ***	0.171 ***	0.108 ***
		S.E.	(0.0101)	(0.0154)	(0.00999)
Southern Guangxi urban agglomeration	EMR-SR-EGR	coefficient	0.232 ***	0.134 ***	0.131 ***
		S.E.	(0.0107)	(0.0221)	(0.00921)
Hohhot-Baotou-Erdos-Yulin urban agglomeration	EMR-SR-EGR	coefficient	0.225	−0.0150	--
		S.E.	--	--	--
Harbin-Changchun urban agglomeration	EMR-SR-EGR	coefficient	0.242 ***	0.154 ***	0.108 ***
		S.E.	(0.0148)	(0.0137)	(0.0118)
Northern Tianshan Mountains urban agglomeration	EMR-SR-EGR	coefficient	1.047	--	--
		S.E.	--	--	--
Ningxia-Yellow River urban agglomeration	EMR-SR-EGR	coefficient	0.302	0.133	--
		S.E.	--	--	--
Shandong Peninsula urban agglomeration	EMR-SR-EGR	coefficient	0.179 ***	0.124 ***	0.102 ***
		S.E.	(0.00711)	(0.0144)	(0.00447)
Chengdu-Chongqing urban agglomeration	EMR-SR-EGR	coefficient	0.190 ***	0.126 ***	0.116 ***
		S.E.	(0.00999)	(0.00556)	(0.00909)
Central Shanxi urban agglomeration	EMR-SR-EGR	coefficient	0.244 ***	0.160 ***	0.116 ***
		S.E.	(0.0179)	(0.00717)	(0.0114)
West Taiwan Straits urban agglomeration	EMR-SR-EGR	coefficient	0.172 ***	0.199 ***	0.111 ***
		S.E.	(0.00991)	(0.0211)	(0.00932)
Central Yunnan urban agglomeration	EMR-SR-EGR	coefficient	0.134	0.268	--
		S.E.	--	--	--
Pearl River Delta urban agglomeration	EMR-SR-EGR	coefficient	0.171 ***	0.138 ***	0.0983 ***
		S.E.	(0.0120)	(0.0138)	(0.00883)
Central and Southern-Liaoning urban agglomeration	EMR-SR-EGR	coefficient	0.229 ***	0.140 ***	0.106 ***
		S.E.	(0.0139)	(0.0140)	(0.00672)
Yangtze River Delta urban agglomeration	EMR-SR-EGR	coefficient	0.170 ***	0.153 ***	0.111 ***
		S.E.	(0.00454)	(0.0101)	(0.00600)
Middle Yangtze urban agglomeration	EMR-SR-EGR	coefficient	0.174 ***	0.151 ***	0.102 ***
		S.E.	(0.00610)	(0.0186)	(0.00510)
Central Guizhou urban agglomeration	EMR-SR-EGR	coefficient	0.211	−0.101	0.216
		S.E.	--	--	--

Note: *** and ** respectively represent significant levels of 1% and 5%.

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
