# Peer review of "The Coupling Coordination Degree of Economic, Social and Ecological Resilience of Urban Agglomerations in China"

_ijerph, 2022, doi:10.3390/ijerph20010413_

Round 1

Reviewer 1 Report

Undoubtedly, this research is interesting. It enriches the study of urban resilience. At the same time, the assessment on the coordination degree of economic, social and ecological resilience has deepened our understanding of the differences between urban agglomerations. However, there are some weaknesses that should be accurately checked and reviewed.

(1) The introduction should specify the innovation points of the article and compare the previous research work.

(2) The introduction needs to strengthen the definition of resilience and supplement the corresponding literature

(3) The logic in Figure 3 is a bit confusing. You may need to build the framework more hierarchically and logically.

(4) b in Figure 6 needs to be adjusted compared with a to be more standardized.

(5) At the end of the introduction, it is mentioned to discuss the interaction feedback mechanism between different dimensions, but you don't seem to have come up with the mechanism. It seems that you just discussed the interaction relationship between them.

(6) The coordination between different dimensions of different urban agglomerations can also be further discussed in the 4.2.

Author Response

Dear Editors,

Thank you for your useful comments, questions and suggestions on the contents and structure of my manuscript. We have modified the manuscript accordingly, and the detailed corrections are listed below point by point:

Undoubtedly, this research is interesting. It enriches the study of urban resilience. At the same time, the assessment on the coordination degree of economic, social and ecological resilience has deepened our understanding of the differences between urban agglomerations. However, there are some weaknesses that should be accurately checked and reviewed.

  • The introduction should specify the innovation points of the article and compare the previous research work.

 Response: Thanks so much for your valuable comments and suggestions. In the introduction, we added the studies on the definition of resilience, and the comparison between the relevant literature.

  • The introduction needs to strengthen the definition of resilience and supplement the corresponding literature.

 Response: Thanks so much for your valuable comments and suggestions. In the revised manuscript, we have added the summary of the predecessors' work and the innovative points of this paper, in line 80.

  • The logic in Figure 3 is a bit confusing. You may need to build the framework more hierarchically and logically.

 Response: Thanks so much for your valuable comments and suggestions. The logical diagram in Figure3 has been modified.

  • b in Figure 6 needs to be adjusted compared with a to be more standardized.

 Response: Thanks so much for your valuable comments and suggestions. Figure 6 was remade. The coordinate system of the three figures was the same, and the size and position of the three figures were as similar as possible.

  • At the end of the introduction, it is mentioned to discuss the interaction feedback mechanism between different dimensions, but you don't seem to have come up with the mechanism. It seems that you just discussed the interaction relationship between them.

 Response: Thanks so much for your valuable comments and suggestions. We found it challenging to understand the interactive feedback mechanism between the three types of resilience, so we replaced the mechanism mentioned in the article with the interactive relationship in the revised draft .

(6) The coordination between different dimensions of different urban agglomerations can also be further discussed in the 4.2.

 Response: Thanks so much for your valuable comments and suggestions. Part 4.2 discusses the coordination degree of three dimensions between different urban agglomerations, and Part 4.3 discusses the coordination degree of two dimensions in detail. Differences between different dimensions of different urban agglomerations are added in the analysis.

All changes have been marked in red.

The authors is greatly thankful to editors and anonymous reviewers for sharing their valuable comments that significantly improved the quality of the paper. If you have any questions, please feel free to contact us. We appreciate your support very much.

Yours sincerely,

December 15th, 2022

Reviewer 2 Report

First of all, the paper addresses a current topic, relevant considering that the phenomenon of urban agglomeration creates increasingly pressing economic, social, environmental and infrastructure problems. 

I believe that the article is developed according to the requirements of a scientific article.

I noticed some minor problems related to the construction of the phrases (Some phrases are too long and hard to follow. Sometimes the verbs are not put in the correct grammatical form (e.g. lines 122-128)) and some drafting errors (e.g. line 9 ”the the coupling…”, line 18, etc.).

As a suggestion, it would be better to provide with an explanation related to the structure of the paper - right from the beginning of the paper (in the Introduction). In addition to stating the objective (which is still presented in the abstract and at the end of the introduction), details can be given about how the paper is structured and how the topic is approached.

Otherwise, the article is well structured, the appropriate methodology for pursuing the proposed objective. The results are interpreted not only statistically, but also from the perspective of the three elements (economic, social and ecological) considered.

Author Response

Dear Editors,

Thank you for your useful comments, questions and suggestions on the contents and structure of my manuscript. We have modified the manuscript accordingly, and the detailed corrections are listed below point by point:

(1) First of all, the paper addresses a current topic, relevant considering that the phenomenon of urban agglomeration creates increasingly pressing economic, social, environmental and infrastructure problems. I believe that the article is developed according to the requirements of a scientific article. I noticed some minor problems related to the construction of the phrases (Some phrases are too long and hard to follow. Sometimes the verbs are not put in the correct grammatical form (e.g. lines 122-128)) and some drafting errors (e.g. line 9 ”the the coupling…”, line 18, etc.).

 Response: Thanks so much for your valuable comments and suggestions. We have asked an English expert with Geography knowledge help us to improve the expression of the manuscript .

(2) As a suggestion, it would be better to provide with an explanation related to the structure of the paper - right from the beginning of the paper (in the Introduction). In addition to stating the objective (which is still presented in the abstract and at the end of the introduction), details can be given about how the paper is structured and how the topic is approached. Otherwise, the article is well structured, the appropriate methodology for pursuing the proposed objective. The results are interpreted not only statistically, but also from the perspective of the three elements (economic, social and ecological) considered.

 Response: Thanks so much for your valuable comments and suggestions. At the end of the introduction, the structure of the paper (98 lines) were introduced, and the following long sentences are modified.

All changes have been marked in red.

The authors is greatly thankful to editors and anonymous reviewers for sharing their valuable comments that significantly improved the quality of the paper. If you have any questions, please feel free to contact us. We appreciate your support very much.

Yours sincerely,

December 15th, 2022

Reviewer 3 Report

The submitted manuscript addresses the three dimensions of resilience in urban agglomerations in China. Economic, social and ecological resilience levels for 2019 have been evaluated using the fuzzy logic method for 19 urban agglomerations. The urban agglomerations are grouped into three categories, i.e., promoting urban agglomerations, developing urban agglomerations, and growing urban agglomerations. The coupling coordination degree of three dimensions has also been evaluated.

Although the topic seems interesting, the present manuscript is not suggested for publication. Language and style jeopardize the comprehension and readability of the manuscript. There are too many very long sentences (L86-92; L235-245), unclear sentences (L148-150: what does which refer to?; L231-233: missing verb), and typos (L9: the the; L19 to contributes; L29 spaces; L106: missing comma; L112: And; L201: unite?). 

Furthermore, although the fuzzy logic and the coupling coordination degree models have been described in detail, the methodology applied in the research needs to be clarified further. Specifically, the selection of the indices is fundamental. However, it is not reported in the manuscript why the authors have selected those specific indices for addressing economic, social, and ecological resilience. Moreover, the authors use twice the generic expression “other indexes” without specifying which indices represent “ecological resources affecting economic resilience” and which represent “environmental protection affecting social resilience”. In Table 1, it is indicated AQI as an index, but the meaning of this acronym is not reported. The selection of appropriate indices is essential for conducting this type of study and determines the appropriateness of the research design, the adequacy of the methodology, and the presentation of the results. These features seem not satisfactorily achieved in the present form of the manuscript.

Author Response

Dear Editors,

Thank you for your useful comments, questions and suggestions on the contents and structure of my manuscript. We have modified the manuscript accordingly, and the detailed corrections are listed below point by point:

The submitted manuscript addresses the three dimensions of resilience in urban agglomerations in China. Economic, social and ecological resilience levels for 2019 have been evaluated using the fuzzy logic method for 19 urban agglomerations. The urban agglomerations are grouped into three categories, i.e., promoting urban agglomerations, developing urban agglomerations, and growing urban agglomerations. The coupling coordination degree of three dimensions has also been evaluated.

  • Although the topic seems interesting, the present manuscript is not suggested for publication. Language and style jeopardize the comprehension and readability of the manuscript. There are too many very long sentences (L86-92; L235-245), unclear sentences (L148-150: what does which refer to?; L231-233: missing verb), and typos (L9: the the; L19 to contributes; L29 spaces; L106: missing comma; L112: And; L201: unite?). 

Response: Thanks so much for your valuable comments and suggestions. Excessively long sentences in L86-92, L235-245 was modified (now L98-104; L244-264), the "which" of L161  and the sentence error of L241 were modified; Delete the "The" In L9, and the comma in L19 were modified. In addition, We have asked an English expert with Geography knowledge help us to improve the expression of the manuscript .

  • Furthermore, although the fuzzy logic and the coupling coordination degree models have been described in detail, the methodology applied in the research needs to be clarified further.

Response: Thanks so much for your valuable comments and suggestions. In the method section, we further explain the function of the algorithm in this paper in detail.

  • Specifically, the selection of the indices is fundamental. However, it is not reported in the manuscript why the authors have selected those specific indices for addressing economic, social, and ecological resilience.

Response: Thanks so much for your valuable comments and suggestions. In “3.3. Index selection”, we discussed in detail the reasons for the selection of these indicators.

  • Moreover, the authors use twice the generic expression “other indexes” without specifying which indices represent “ecological resources affecting economic resilience” and which represent “environmental protection affecting social resilience”. In Table 1, it is indicated AQI as an index, but the meaning of this acronym is not reported. The selection of appropriate indices is essential for conducting this type of study and determines the appropriateness of the research design, the adequacy of the methodology, and the presentation of the results. These features seem not satisfactorily achieved in the present form of the manuscript.

Response: Thanks so much for your valuable comments and suggestions. In Table 1, the abbreviation AQI is changed to Air Quality Index, the reasons for the indicators selection are elaborated in more detail, and We also optimized the whole article in the manuscript. 

All changes have been marked in red.

Thanks so much for your valuable comment. we would like to express our sincere thanks to you for your most valuable comments and suggestions in improving this paper greatly.

Yours sincerely,

December 15th, 2022